# Rapid Pyrolysis of SmBa$_2$Cu$_3$O$_{7-\delta}$ Films in CSD-MOD Using Extremely-Low-Fluorine Solutions

**Minjuan Li [1], Pablo Cayado [2],\* , Manuela Erbe [2], Alexandra Jung [2], Jens Hänisch [2], Bernhard Holzapfel [2], Zhiyong Liu [1] and Chuanbing Cai [1]**

[1] Shanghai Key Laboratory of High Temperature Superconductors, Physics Department, Shanghai University, Shanghai 200444, China; lmj102@126.com (M.L.); zyliu@shu.edu.cn (Z.L.); cbcai@shu.edu.cn (C.C.)

[2] Karlsruhe Institute of Technology (KIT), Institute for Technical Physics (ITEP), Hermann-von-Helmholtz-Platz 1, 76344 Eggenstein-Leopoldshafen, Germany; manuela.erbe@kit.edu (M.E.); alexandra.jung@kit.edu (A.J.); jens.haenisch@kit.edu (J.H.); bernhard.holzapfel@kit.edu (B.H.)

\* Correspondence: pablo.cayado@kit.edu

**Abstract:** SmBa$_2$Cu$_3$O$_{7-\delta}$ (SmBCO) films have been prepared by chemical solution deposition starting from extremely-low-fluorine solutions (7% fluorine with respect to standard full trifluoroacetate solutions). Smooth and homogeneous SmBCO films could be achieved at heating rates of up to 20 °C/min during pyrolysis. The best films were achieved at a crystallization temperature of 810 °C and 50 ppm of oxygen partial pressure. At these conditions, the ~270 nm thick SmBCO films grow mostly *c*-axis-oriented with $J_c^{sf}$ values at 77 K of ~2 MA/cm$^2$ and critical temperatures $T_c$ of up to 95.0 K. These results demonstrate that using extremely-low-fluorine solutions is very attractive since the production rate can be largely increased due to the solutions' robustness during pyrolysis retaining a remarkable quality of the grown films. Nevertheless, further optimization of the growth process is needed to improve the superconducting properties of the films.

**Keywords:** SmBCO; chemical solution deposition; rapid pyrolysis; metal-organic deposition; extremely low fluorine

## 1. Introduction

The second-generation (2G) superconducting tapes or coated conductors (CCs) have promising perspectives for a number of applications such as motors, generators or fault-current limiters. The CCs are based on high-temperature superconductors, in particular *RE*Ba$_2$Cu$_3$O$_{7-\delta}$ (*RE*BCO, *RE* = rare earth) compounds. These are compounds with a great potential due to their high values of critical temperature ($T_c$), upper critical field and irreversibility field, as well as the associated large current-carrying capability in applied magnetic fields [1–3].

Among various techniques to prepare *RE*BCO films, the low-cost and easy-to-scale chemical solution deposition (CSD) is an interesting approach for the industrial production of *RE*BCO films and CCs [4–9]. Most previous studies follow the well-known TFA-MOD route [10]. During the pyrolysis, the fluorine-containing precursor solutions decompose to metal fluorides avoiding the formation of BaCO$_3$, which would be difficult to decompose during the crystallization stage [11–14]. Yet, TFA-MOD needs long pyrolysis times to obtain defect-free pyrolyzed films [14–16] and, thus, process shortening is required to allow for large production rates.

In order to overcome this limitation, several advanced MOD methods have been developed in recent years [17–21]. In our previous work [22], various precursor solutions with different fluorine contents have been tested, including the conventional low-fluorine (CLF) solution with a fluorine content of 54% (F-54%), the super low-fluorine (SLF) solution (F-31%), and the extremely-low-fluorine

(ELF) solution (F-7%) with respect to the full trifluoroacetates (TFA) solution (F-100%). As shown by Li et al. [22], the heating rate of the pyrolysis step for YBCO films can be increased as the fluorine content of the precursor solution decreases. However, the reduction of the fluorine content in the precursor solution has other advantages, too: It is environmentally friendly and can also improve the homogeneity of the films during the pyrolysis process. Using ELF precursor solutions, excellent superconducting properties have been achieved in YBCO films with much faster heating rates than with other types of solutions [20,22]. However, only little work has been spent to grow *RE*BCO films with other rare earth elements with ELF solutions.

As pointed out by Wu et al. [20] and Li et al. [22], the reaction mechanism for the formation of *RE*BCO films using ELF solutions is still a "BaF$_2$" process, which is a well-understood process. There are also multiple studies on the influence of the $RE^{3+}$ ionic radius on the superconducting properties of *RE*BCO compounds, e.g., [23–27], and some of these compounds have been shown to exhibit larger $T_c$ and $J_c$ values than YBCO [28–30]. However, the synthesis of some of these compounds is much more complicated than for YBCO and requires more effort for both optimization and understanding. On the one hand, large *RE* ions, like $Nd^{3+}$ or $Sm^{3+}$, tend to partially substitute the $Ba^{2+}$ ions and, on the other hand, small $RE^{3+}$ ions, such as $Yb^{3+}$ and $Lu^{3+}$, do not fit properly in their lattice site introducing vacancies. Both facts cause a drastic decrease of the *RE*BCO phase stability [31–33]. In particular, the synthesis of SmBCO is one of the most challenging among the *RE*BCO phases because the tendency of $Sm^{3+}$ ions to substitute for $Ba^{2+}$ is one of the highest among the $RE^{3+}$ ions [28]. However, such films, properly prepared, tend to present higher $T_c$ values and irreversibility fields than YBCO as well as a smaller field dependence of the critical current density $J_c(T, B, \theta)$ [34–37].

In this study, we have prepared pristine SmBCO films using ELF solutions for the first time. The main objective was to explore the possibilities of using SmBCO-ELF solutions to reduce the total time of the pyrolysis process with fast heating ramps for obtaining pyrolyzed films without inhomogeneities. The reduction of the pyrolysis time is a key point for the economical use of the CSD process in long-length processes. After investigating the influence of the heating rate during pyrolysis on the homogeneity of the films, we have made a study of the crystallization process focusing in the influence of crystallization temperature ($T_{crys}$) and oxygen partial pressure ($pO_2$) on the structure and superconducting properties of the final SmBCO films, to determine the most appropriate crystallization process for this kind of films.

## 2. Experimental Details

### 2.1. Solution and Thin Film Preparation

The preparation of the SmBCO-ELF solution consists of several steps. First, Sm, Ba, and Cu acetates (purity > 99.99%, Alfa Aesar, Kandel, RLP, Germany) are weighed out in the stoichiometric ratio of 1:2:3 for the metal cations. Then, 1/3 of the mass of the Ba acetate is dissolved in trifluoroacetic acid and deionized water to convert it to trifluoroacetate while the rest of the acetates is dissolved in propionic acid (99%). Both solutions are dried by a rotary evaporator resulting in highly viscous residues that are re-diluted in anhydrous methanol (99.9%). In order to reduce the undesired residual water and other impurities, the methanol is evaporated again three times. Finally, after a last re-dilution in methanol, the two remaining solutions are mixed, and the final concentration of 2 mol·L$^{-1}$ (sum of total metal concentration) is adjusted by adding or evaporating methanol. With this procedure, a ratio F/Ba = 2 is achieved, which is the minimum amount of F necessary to allow a full conversion of the Ba precursor to BaF$_2$ avoiding the formation of BaCO$_3$ [12].

For the preparation of the full-TFA solution (SmBCO-TFA) we used the procedure described in our previous works [17,27]. In summary, this is a 1.5 molar solution (over all cations) with a 1:2:3 ratio of Sm, Ba, and Cu dissolved in dry methanol.

Both precursor solutions, SmBCO-TFA and SmBCO-ELF, were deposited on (001)-oriented $LaAlO_3$ (LAO) single crystal substrates via spin coating (6000 rpm, 2 min). The as-deposited films were thermally treated in a tubular furnace. For the SmBCO-ELF films, the pyrolysis was performed in a single-step thermal process at 500 °C, which was kept for 30 min in humidified pure oxygen with a dew point of 25 °C. Several heating ramps from 5 °C/min to 20 °C/min were tested. For the SmBCO-TFA films we used the "standard pyrolysis process" described by Erbe et al. [17]. Subsequently, the SmBCO-ELF films were crystallized using a similar growth process to the one described by Cayado et al. [27]. In this particular case, the films were heated for 90 min in a humid $O_2/N_2$ gas at several $T_{crys}$ (780–840 °C) and $pO_2$ (20–100 ppm). The following oxygenation process was carried out at 450 °C for 2 h.

## 2.2. Thin Film Characterization

Microstructure and phase purity of the ~270 nm films were investigated by X-ray diffraction (XRD) using a *Bruker* D8 diffractometer with CuK$\alpha$ radiation. The surface morphology of the pyrolyzed films was analyzed by a Keyence VHX-1000 digital microscope with motorized *z*-axis while the surface morphology and the cross-sections of the grown films were analyzed by a LEO 1530 scanning electron microscope (SEM) with field emission gun by Zeiss and by a Bruker Dimension Edge atomic force microscope (AFM). The AFM images have also been used for the thickness measurements by measuring the step between film surface and substrate after etching some areas of the films with nitric acid. Self-field $J_c$ at 77 K, $J_c^{sf}$, was measured inductively with a Cryoscan (*Theva*, 50 μV criterion). $T_c$ ($T_{c,90}$, i.e., the temperature at which the resistance is 90% of the value above the transition) and $T_c$ ($T_{c,90} - T_{c,10}$) were determined from resistivity-temperature curves measured with a 14-T Quantum Design Physical Property Measurement System (PPMS) in four-point method.

## 3. Results and Discussion

### 3.1. Influence of the Heating Rate during Pyrolysis

The thermal profile of the pyrolysis had to be newly designed for the SmBCO-ELF films since no previous results were reported for this particular type of films and the temperature ramp towards the pyrolysis dwell temperature is one of the decisive parameters for good film growth [38,39]. Heating rates of 5, 10, 15, and 20 °C/min were tested for their influence on the surface morphology for both SmBCO-ELF and SmBCO-TFA films.

With ELF solutions, homogeneous, smooth, and defect-free films were achieved even with the highest heating rate of 20 °C/min (Figure 1). For TFA films, on the other hand, even the lowest ramp of 5 °C/min led to an inhomogeneous surface with buckling, and higher heating rates resulted in worse morphologies, yet. This is a very remarkable result, since the pyrolysis can be drastically shortened by the use of ELF solutions. As shown in the experimental section, the used ELF solutions are free of any extra additive often used to increase the elasticity of the films during pyrolysis and avoid the formation of inhomogeneities. The use of such high heating rates during pyrolysis with an additive-free solution has no precedent in literature.

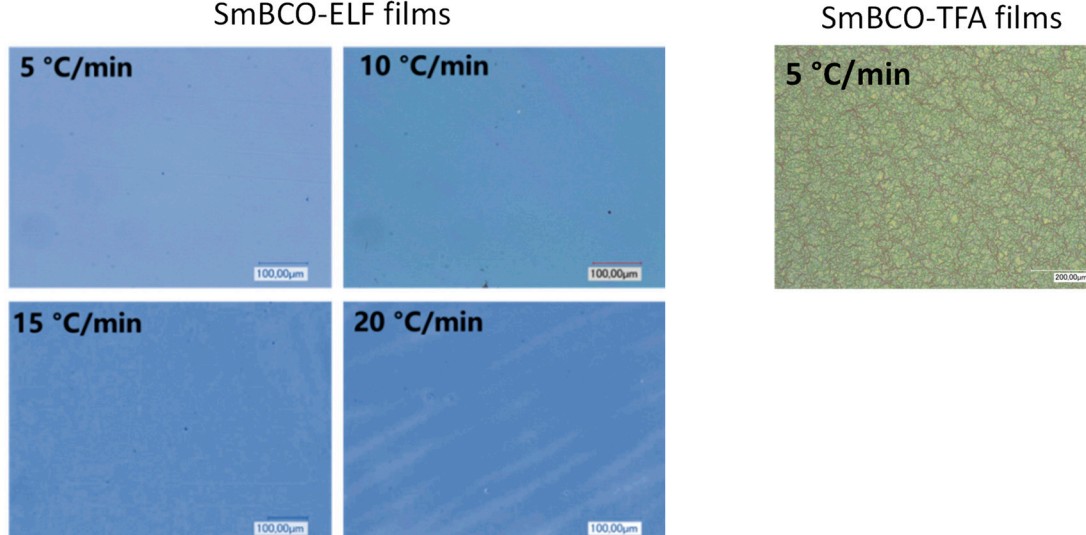

**Figure 1.** Surface morphology of pyrolyzed SmBCO-ELF and SmBCO-TFA films at different heating rates observed in optical microscope images taken in reflection mode with white illumination and without polarization filters.

### 3.2. Optimization of the Growth Process

The SmBCO phase is formed for all explored $T_{crys}$ with a significant preference of the SmBCO (00$l$) orientation in all films (Figure 2a). However, additional reflections of SmBCO (103) and SmBCO (108) are observed, which are associated to randomly oriented grains. In order to estimate the amount of misoriented grains (represented by the (103) SmBCO peak) formed in comparison with the epitaxial fraction (represented in this case by the (005) SmBCO peak), one can use the following expression (Equation (1)):

$$\frac{\text{I}(103)\text{SmBCO}}{\text{I}(103)\text{SmBCO} + \text{I}(005)\text{SmBCO}} \tag{1}$$

Figure 2b shows the evolution of the ratio (103)/(005) after applying Equation (1). One can conclude that the intensities of the reflections associated to misoriented grains increase both at low and high $T_{crys}$ reaching a minimum at 810 °C. Secondary phases such as $Sm_2O_3$ are found in the films as well. The amount of these secondary phases is also larger at low and high temperatures being almost negligible between 800 and 820 °C. Thus, temperatures of 800–820 °C are the most suitable $T_{crys}$ window for SmBCO-ELF films with respect to both phase purity and crystallographic orientation.

The increase of the $pO_2$ up to 100 ppm at a constant $T_{crys} = 810$ °C reduce the formation of misoriented grains but enhances the formation of $a$-axis grains enormously (marked with + in Figure 2c). Moreover, the intensities of the reflections associated to (00$l$)-oriented grains decrease. This enhancement of $a$-axis grains with $pO_2$ is common also in other REBCO compounds and was also observed previously in the particular case of SmBCO-TFA films [38]. At 20 and 50 ppm, the films show rather similar characteristics but with slightly lower intensities of the (00$l$) reflections at the lowest partial pressure of oxygen, Figure 2d). Therefore, 50 ppm is the optimal $pO_2$ at 810 °C.

In general, as it was observed for YBCO-ELF, the films grown from ELF solutions tend to form more misoriented grains than films grown from TFA or conventional low-fluorine solutions [22]. This means than for this type of solutions the supersaturation values that allows the fully epitaxial growth are more difficult to reach and it is necessary a more carefully optimization of the growth parameters. SmBCO-ELF films show a similar trend towards texture deterioration and probably even more due to the difficulties of the synthesis of the SmBCO phase itself that, as mentioned before, come from the great tendency of $Sm^{3+}$ ion to partially substitute the $Ba^{2+}$ ions.

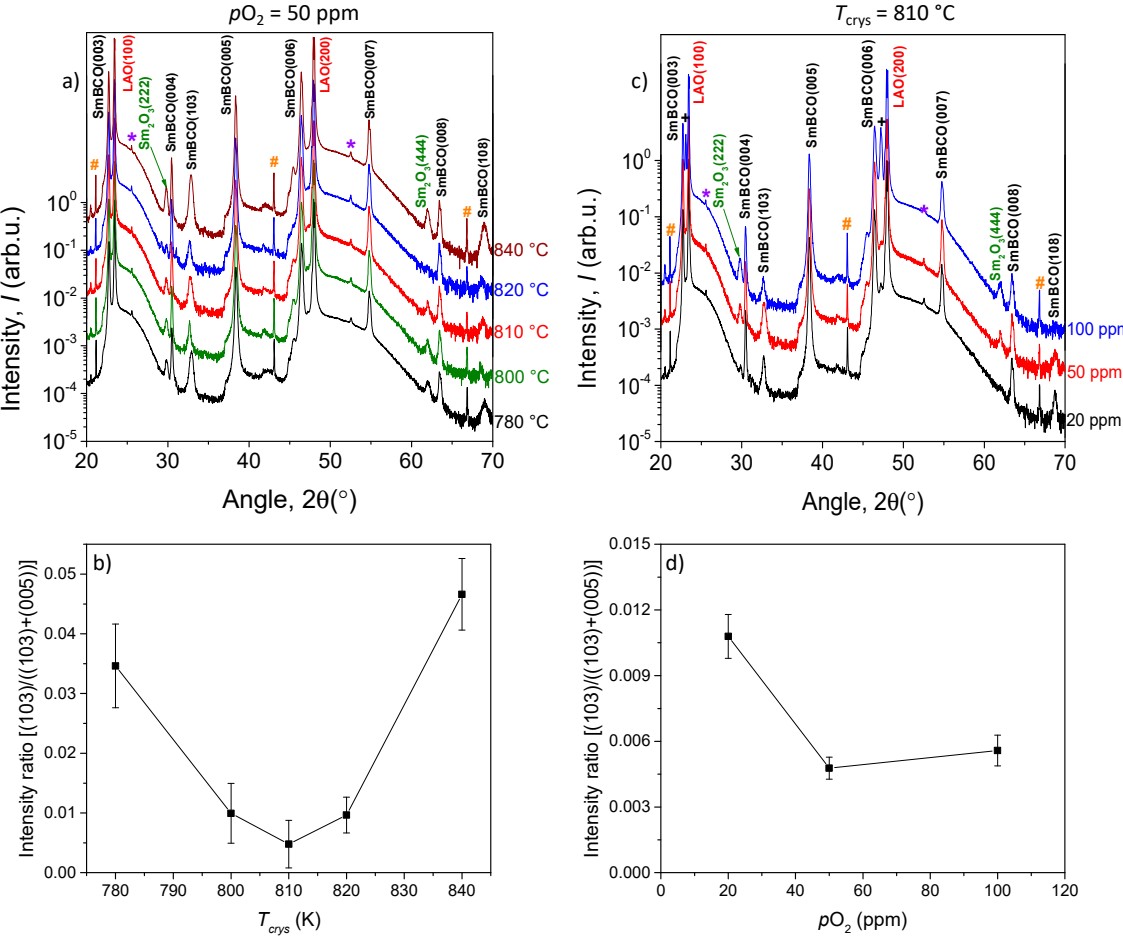

**Figure 2.** XRD patterns of SmBCO films grown (**a**) at different crystallization temperatures and $pO_2$ = 50 ppm and (**c**) at different oxygen partial pressures and $T_{crys}$ = 810 °C. The evolution of the (103)BCO/(005)SmBCO peak intensity ratio calculated using Equation (1) with $T_{crys}$ and $pO_2$ for is shown in (**b,d**), respectively. The intensity ratio of the peaks marked with # and * come from the experimental setup.

The tendencies found in XRD are, to some extent, reflected in the surface morphology observed in SEM (Figure 3). Clearly, the areal density of randomly oriented grains (needle-shaped structures oriented in any possible direction) increases at low temperatures and low values of $pO_2$. Increasing the $pO_2$ up to 100 ppm results in an enhancement of the areal density of *a-b* grains (needle-shaped structures with orthogonal orientation to each other). At high temperatures, the films present less misoriented grains at the surface, but they seem to be embedded in the films and with a much larger size (white arrows in Figure 3, 840 °C). In general, the films appear very dense.

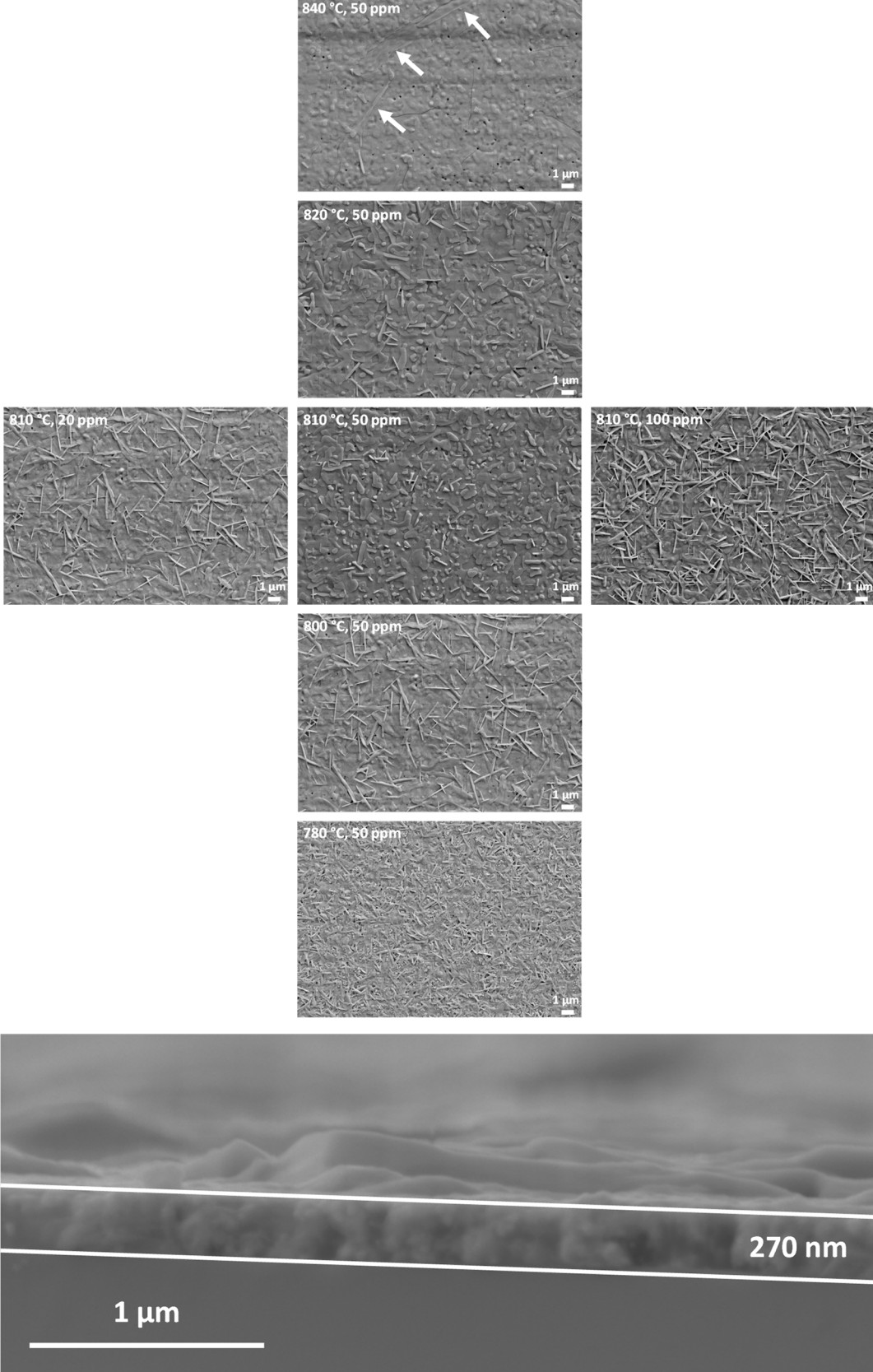

**Figure 3.** Surface morphology of SmBCO-ELF films depending on crystallization temperatures and $pO_2$ as observed in SEM. At the bottom of the figure a cross-section SEM image is displayed showing that the thickness of the films is ~270 nm.

The AFM measurements carried out on the film grown at 810 °C and 50 ppm (Figure 4) present similar features as the SEM image shown before. Randomly oriented grains are visible at the surface of the film (Figure 4a)). Additionally, these AFM measurements allowed to confirm that the thickness of the films is ~270 ± 30 nm.

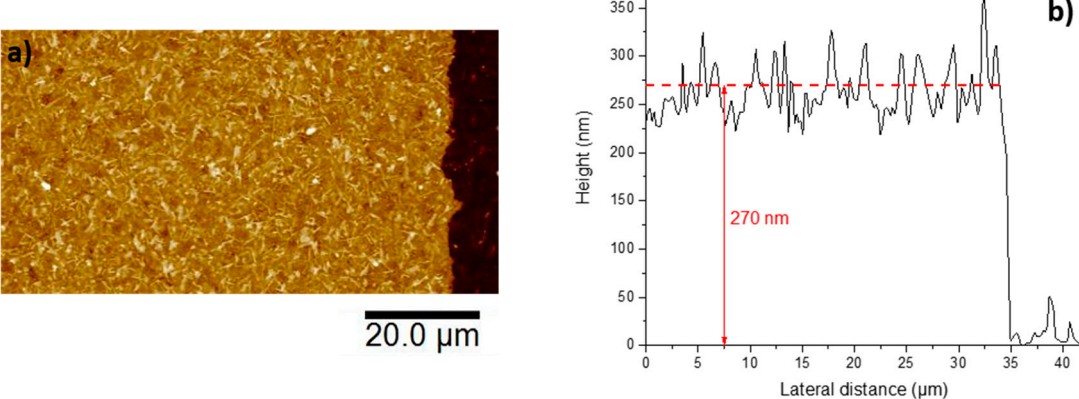

**Figure 4.** Surface morphology (**a**) and thickness of the films (**b**) obtained from AFM measurements carried out on the SmBCO film grown at at 810 °C and 50 ppm.

The trends of $J_c^{sf}$ at 77 K (Figure 5a) with $T_{crys}$ and $pO_2$ are in agreement with the structural observations discussed above. Films grown at 50 ppm show higher $J_c^{sf}$ values than the films grown at 20 or 100 ppm. This is due to the formation of multiple misoriented grains, see Figures 2 and 3, at these $pO_2$ values. The highest $J_c^{sf}$ values are reached at 810 °C (820 °C for 100 ppm), which is again attributed to larger amounts of misoriented grains at lower or higher temperatures. At the best growth conditions $T_{crys}$ = 810 °C and $pO_2$ = 50 ppm, $J_c^{sf}$ at 77 K reaches 2.1 MA cm$^{-2}$ with a $T_c$ of 95.0 K and a transition width $T_c$ of 0.9 K (Figure 4b). To our knowledge, this $T_c$ value is the highest reported for SmBCO films deposited either on single crystals or on metallic tapes [35,37,39,40]. Additionally, the $J_c^{sf}$ values at 77 K are among the highest reported for CSD-grown SmBCO film [39–41].

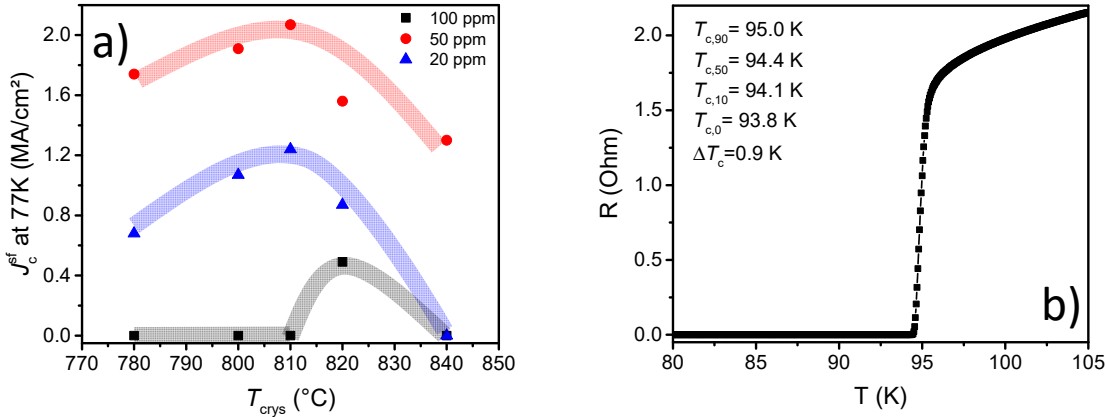

**Figure 5.** (**a**) Dependence of the $J_c^{sf}$ at 77 K on $T_{crys}$ and $p(O_2)$ and (**b**) $R(T)$ curve of a SmBCO-ELF film grown at 810 °C and 50 ppm.

## 4. Conclusions

SmBCO films were prepared by CSD with extremely-low-fluorine (ELF) solutions on LAO substrates. This type of solutions is much more robust during pyrolysis than TFA solutions and, consequently, the deposition of homogenous and defect-free pyrolyzed SmBCO films is much more reproducible than with TFA solutions, even for extremely fast heating rates (20 °C/min). The films present a certain tendency to generate misoriented grains during the crystallization with a minimum

at a crystallization temperature of 810 °C and 50 ppm of oxygen partial pressure. The ~270 nm SmBCO-ELF films obtained at these conditions present $J_c^{sf}$ values at 77 K up to 2.1 MA cm$^{-2}$ and $T_c$ values up to 95.0 K with a $T_c$ of 0.9 K. These results show the promising perspectives of the ELF solutions for the preparation of high-quality SmBCO films. However, more effort is needed to clarify why this type of solutions tends to generate more misoriented grains than other solutions and to optimize the growth process to procure fully epitaxial films. This will allow a further improvement of the superconducting properties, especially $J_c$, which will make this type of solutions and films even more attractive for their use in the long-length CC production.

**Author Contributions:** Conceptualization, M.L., P.C., M.E., J.H. and B.H.; Data curation, M.L. and P.C.; Formal analysis, M.L., P.C. and A.J.; Investigation, M.L. and P.C.; Methodology, M.L., P.C. and A.J.; Project administration, P.C., M.E., A.J., J.H. and Z.L.; Resources, M.L., P.C. and M.E.; Supervision, P.C., M.E., J.H., B.H., Z.L. and C.C.; Validation, M.L. and P.C.; Visualization, M.L.; Writing—original draft, M.L. and P.C.; Writing—review & editing, M.L., P.C., M.E., A.J., J.H., B.H., Z.L. and C.C. All authors have read and agreed to the published version of the manuscript.

**Funding:** This work was supported in part by the Science and Technology Commission of Shanghai Municipality (16521108402, 13111102300, and 14521102800), the National Natural Science Foundation of China (51572165), and National Key Research and Development Program (2016YFF0101701).

**Conflicts of Interest:** The authors declare no conflict of interest.

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
