# Peer review of "Rapid Pyrolysis of SmBa2Cu3O7-δ Films in CSD-MOD Using Extremely-Low-Fluorine Solutions"

_coatings, doi:10.3390/coatings10010031_

Round 1
Reviewer 1 Report
The paper is interesting and well written.
Experimental findings support the main conclusions drawn by the authors
The topic is well suited for Coatings
The paper can be published in the present form
Reviewer 2 Report
In their manuscript "Rapid Pyrolysis of SmBa2Cu3O7-delta Films in CSD-MOD ..." Pablo Cayado and coworkers investigate thin SmBCO films prepared from extremely-low-fluorine solutions with respect to, e.g., their structure, morphology, and superconducting properties. Even so the underlying mechanism is not fully understood, the found process window of fast heating rates for high-quality films is an important find for possible applications. Therefore, these results will prove to be of importance for thin film community.
However, before the paper is ready for publication, the authors should address some points.
1. Page 2, line 91: here the authors use K/min as unit for the heating rates. At other places they use °C/min. Please stick to one unit.
2. Figure 1: The authors should provide information of what type of microscope images are shown. Transmission or reflection? White light illumination? Any polarization filters involved? What is the reason for the bright, diagonal running stripes in the 20 °C/min image?
3. page 4, line 129: the authors claim that the films grow epitaxially, but don't provide any evidence for that.
4. Figure 2: no error bars are shown in b) and d). Without error bars these two figures are useless.
5. page 5, line 159: It is stated that "needle-shaped structures [are] oriented in any possible direction". However, the authors do not provide any proof that this really is the case. An analysis of the distribution of needle directions should be added or, at least, the result of such an analysis should be provided.
6. Figure 4: similar to Figure 2, no error bars are provided.
7. For some of the references the journal names are abbreviated, for some they are not.
Reviewer 3 Report
This paper reports the rapid pyrolysis of SmBa2Cu3O7-δ films in CSD-MOD using extremely-low-fluorine solutions. The paper is well written and structured. There are ertain issues that need to be addressed befor publication. Hence, I want the authors to provide the major revision of the following comments:
1- What is the novelty of this study? Explain the contribution of this study to science in the introduction?
2- Line 27: Name a few of the number of applications for clarity.
3- Figure 2: It is preferred to have some gap in the adjoining XRD peaks for clarity. It is better if the authors redraw the peaks.
4- What was the film thickness? Did the different temperatures affect the film thickness/ Provide the cross-sectional SEM images of the film for clarity. Also, provide the plain-view magnified images of the individual films.
5- Page 6, Line 170: Authenticate the reported fact, "This is due to the formation of multiple misoriented grains at these pO2 values." with a bibliographic reference.
6- The authors have reported only the growth of the films. To what applications do the film be useful in science and technology. Did the authors perform the superconductivity film testing?
Round 2
Reviewer 2 Report
Since the autors responded satisfactorily to all of my requests, I think the paper is ready for publication now.
Reviewer 3 Report
The information in SEM images are very important and tricky. I have serious concerns with the film morphology explained in the manuscript and the morphology actually presented in the SEM images.
At first, if the authors have already taken the AFM images then I prefer adding the AFM data into the manuscript for clarity.
Second, the high magnification images are one of the most important information used to qualify the reported films morphology. The most formal and widely accepted trend is to provide a low magnification image, as provided in Figure 3, to confirm the uniformity of the film on the substrate surface. On the contrary, high magnification films are important to qualify the reported morphology of the films. Such as, the authors have reported the formation of needle like structures in the manuscript, however, it is difficult to recognize the reported fact with the provided low magnification images. Hence, it is important to provide both high and low magnification SEM images to provide the complete information to the readers.
Third, the scale bar in the plain view images are too small to be identified. Increase the scale bar dimensions for clarity.
Four, clearly mention the sample from which the cross-sectional image was taken? Normally, the authors provide the cross-sectional images of all the samples to give an evidence that the films had same thickness, however, the authors have not provided the sufficient reference besides providing a single image and mere reporting the fact that the films were 270 nm thick.
Five, the cross-section does not show a single bit of needle like structures as reported in the plain-view SEM images. How would the authors qualify the fact that the films constitute needle-like structures when the cross-section indicates a plain epitaxial film with no needles? The results are self conflicting.
Round 3
Reviewer 3 Report
I accept the paper in present form.